# Recurrent Herpes Simplex Virus Type 1 (HSV-1) Infection Modulates Neuronal Aging Marks in In Vitro and In Vivo Models

**DOI:** 10.3390/ijms22126279

**Published:** 2021-06-11

**Authors:** Giorgia Napoletani, Virginia Protto, Maria Elena Marcocci, Lucia Nencioni, Anna Teresa Palamara, Giovanna De Chiara

**Affiliations:** 1Department of Public Health and Infectious Diseases, Sapienza University of Rome, Laboratory Affiliated to Istituto Pasteur Italia–Fondazione Cenci Bolognetti, 00185 Rome, Italy; giorgia.napoletani@uniroma1.it (G.N.); virginia.protto@uniroma1.it (V.P.); mariaelena.marcocci@uniroma1.it (M.E.M.); lucia.nencioni@uniroma1.it (L.N.); annateresa.palamara@uniroma1.it (A.T.P.); 2Department of Infectious Diseases, Istituto Superiore di Sanità, 00161 Rome, Italy; 3Institute of Translational Pharmacology, National Research Council (CNR), 00133 Rome, Italy

**Keywords:** HSV-1, Herpes simplex virus, recurrent infection, neuronal aging, histone modifications

## Abstract

Herpes simplex virus 1 (HSV-1) is a widespread neurotropic virus establishing a life-long latent infection in neurons with periodic reactivations. Recent studies linked HSV-1 to neurodegenerative processes related to age-related disorders such as Alzheimer’s disease. Here, we explored whether recurrent HSV-1 infection might accelerate aging in neurons, focusing on peculiar marks of aged cells, such as the increase in histone H4 lysine (K) 16 acetylation (ac) (H4K16ac); the decrease of H3K56ac, and the modified expression of Sin3/HDAC1 and HIRA proteins. By exploiting both in vitro and in vivo models of recurrent HSV-1 infection, we found a significant increase in H4K16ac, Sin3, and HDAC1 levels, suggesting that the neuronal response to virus latency and reactivation includes the upregulation of these aging markers. On the contrary, we found a significant decrease in H3K56ac that was specifically linked to viral reactivation and apparently not related to aging-related markers. A complex modulation of HIRA expression and localization was found in the brain from HSV-1 infected mice suggesting a specific role of this protein in viral latency and reactivation. Overall, our results pointed out novel molecular mechanisms through which recurrent HSV-1 infection may affect neuronal aging, likely contributing to neurodegeneration.

## 1. Introduction

HSV-1 is a neurotropic virus able to establish a life-long infection in neurons. After a primary infection in the epithelial cells, the virus contacts the axonal endings of sensory and sympathetic neurons in the site of primary infection [1,2] and, via retrograde axonal transport, reaches the cell bodies in the ganglia of the peripheral nervous system (PNS) where it establishes a latent infection. In this condition, viral replication is blocked, and viral genome remains in the episomal form in the nuclei of infected neurons. Throughout the host’s life, the virus may reactivate as a result of several stimuli, including exposure to UV-light, stress, fever, and immune suppression [2]. Successful reactivation allows the production of new viral particles that, traveling along the axon, return to the primary infection site (e.g., oral mucosal epithelial cells) and give rise to recurrent infection (symptomatic or asymptomatic). More rarely, these viral particles may also reach the central nervous system (CNS) where they can infect neurons causing a severe herpetic encephalitis or, likely, establishing a latent infection [1,3,4,5]. An increasing body of data pointed out herpesviruses infections, such as HSV-1, as co-risk factors in the CNS neurodegenerative diseases, including Alzheimer’s disease (AD), the main form of dementia in the elderly [1,6,7,8,9,10,11,12,13,14,15]. In this context, recent data from our group demonstrated that recurrent HSV-1 infection causes the accumulation of AD hallmarks in mice, including amyloid beta peptides (Aβs) and hyperphosphorylated tau, as well as signs of neuroinflammation and irreversible cognitive deficits [3]. Specifically, we observed a correlation between the number of viral reactivations and the progressive accumulation of these markers in HSV-1-infected mice compared to control ones.

Aging is considered one of the main risk factors for several neurodegenerative disorders, including AD [16,17]. Consistently, some of the major AD players are strictly associated to aging markers; for instance, Aβ oligomers are reported to induce a senescent phenotype in neural stem cells (NSCs), including enlarged and flattened morphology, increased levels of senescence-associated β-galactosidase (SA-β-gal) and activation of cyclin-dependent inhibitor p16^INK4^. These signs have been also identified in both aged cells [18] and AD cortical brain tissues [19,20,21,22,23,24]. Intriguingly, aging may also affect the chromatin landscape: several factors and their post-transcriptional modifications (PTMs) have been indeed pointed out as aging markers, including the acetylation levels of two specific residues of histone H3 and H4, lysine 56 (K56) and lysine 16 (K16) (H3K56ac and H4K16ac). Specifically, a decrease in H3K56ac during replicative senescence was reported, suggesting a role for this PTM in lifespan regulation [21,25]. On the contrary, the acetylation of H4K16 was reported to increase with age, and H4K16ac was found enriched at promoter regions of genes expressed in senescent cells independently of their expression [25,26,27]. The acetylation level of these lysine residues is regulated by the class I histone deacetylases (HDACs) family, which are enzymes involved in transcription and gene repression [28]. For instance, HDAC1 is recruited by several chromatin regulator complexes, such as CoREST/LSD1/REST and Sin3/HDAC1 co-repressor complex [29], two complexes that exert a role in the regulation of HSV-1 replication, repressing viral gene expression. Interestingly, HDAC1 aberrant activity has been linked to neurodegeneration and its export to the cytoplasm promotes axonal and neuronal degeneration in different pathological conditions [30,31]. Moreover, HDAC1 expression increases with aging in the human brain, potentially leading to a neurotoxic effect and, ultimately, to neuronal death [30,32].

The histone regulator A (HIRA) chaperone, involved in DNA synthesis-independent nucleosome assembly through the incorporation of H3.3 histone variant [33], is another crucial player in aging. In aged cells, HIRA participates to the formation of the senescence-associated heterochromatic foci (SAHFs) [34,35] and HIRA complex colocalizes with the tumor suppressor promyelocytic leukemia (PML) protein in PML-nuclear bodies (PML-NBs) [36,37], whose formation depends on cellular stress such as DNA-damage, and oxidative stress. It is noteworthy that HIRA, together with PML, works as a restriction factor for DNA viruses, including HSV-1, e.g., it activates innate cell responses (including Interferon stimulated gene expression) and contributes to viral gene silencing to limit virus replication [38,39,40,41].

In the last few years, many pieces of evidence linked aging to persistent viral infections, including that caused by human immunodeficiency (HIV) and hepatitis C viruses (HCV) [42,43,44] and very recently HSV-1 [45]. Regarding HSV-1, notwithstanding the amount of data on the mechanisms regulating its latency and reactivation in neurons, the long-term effects of recurrent infection on aging are poorly characterized, and the possibility that this infection may also impair and accelerate the normal neuronal aging remains unexplored thus far. Thus, in the present study, we investigated the influence of HSV-1 infections on the host age-related chromatin balance in neurons by evaluating changes in the levels of peculiar markers of aging and associated factors. We exploited both in vitro and in vivo models of recurrent HSV-1 infection in order to study the effect of infection in neurons and its progresses during aging and multiple reactivations [3,46]. We found: (i) an HSV-1-reactivation-dependent decrease of H3K56ac; (ii) an H4K16ac increase and a significant increase of Sin3 and HDAC1 protein persistent within the recurrent HSV-1 infection; (iii) a significant increase in HIRA protein levels during viral latency and its different localization in cortical neurons from HSV-1 infected mouse brains. Overall, our data highlight an additional mechanism through which recurrent HSV-1 infection causes long-term alterations in neurons, thus contributing to neurodegeneration.

## 2. Results

### 2.1. H3K56 and H4K16 Acetylation Levels Are Inversely Modulated by Recurrent HSV-1 Infection in Neurons

To investigate whether recurrent HSV-1 infection may accelerate the normal neuronal aging, we first set up an in vitro model of HSV-1 infection and reactivation in cultures of rat primary neurons as previously described [46] with minor modifications. Briefly, day in vitro 7 (DIV7) primary neurons were inoculated with 0.1 multiplicity of infection (M.O.I.) of HSV-1 or Mock solution (hereafter named HSV-1 and CTRL, respectively) and harvested 24 hours (h) post-infection (p.i.) or 24 h post-viral reactivation (p.r.). The latter group of neurons was treated with 50 µM acycloguanosine (ACV) starting from 18 h before the virus or mock inoculation (Figure 1a) to allow the establishment of latent infection; 10 days post-infection (d.p.i.), neurons were subjected to thermal stress (TS; 10 min, 42 °C) in the absence of ACV to induce HSV-1 reactivation, as described in Materials and Methods. Neuronal aging markers were studied at two experimental time points, resembling two faces of HSV-1 infection in neurons: (i) 24 h p.i. (i.e., when the virus has completed its life-cycle following the primary infection), to study the effects of the virus after the acute infection; (ii) 24 h p.r. (i.e., when it has completed a cycle of latency and reactivation), to study the effects of the virus long-lasting infection following a reactivation cycle after 10 days of latency. Latency establishment, as well as the efficacy of virus infection and reactivation, were assessed by Standard Plaque Assay (SPA) (Figure 1b or In Cell Western (ICW) assay in Appendix A). To assess if acute or recurrent HSV-1 infections could accelerate aging of neurons we focused our investigation on the two lysine residues, H4K16 and H3K56, whose acetylation was reported to be modulated during aging. In particular, it is known that the acetylation levels of H3K56 decrease during aging in various organisms [25,47], whereas those of H4K16 increase [26,27]. Results from Western Blotting (WB) analyses of acetylation levels show a trend increase in H4K16 acetylation (H4K16ac) after primary infection compared to Mock-infected neurons, which was maintained 24 h p.r. (Figure 1c). On the contrary, the acetylation of H3K56, which did not show any alteration 24 h p.i., was significantly reduced after 1 cycle of latency and reactivation in HSV-1-infected neurons with respect to matched controls, (* *p* < 0.05) (Figure 1c), likely indicating that the virus is able to affect the acetylated status of this lysine residue within a long-lasting infection. Overall, these data suggest that recurrent HSV-1 infection could affect the acetylation of these amino acidic residues, especially for H3K56.

### 2.2. Sin3 and HDAC1 Protein Levels Increase during In Vitro Recurrent HSV-1 Infection

To gain insight into the decrease in H3K56ac, we investigated whether HSV-1 infection and reactivation could affect the expression levels of the components of the repressor complex Sin3a/HDAC1, known to be involved in histone deacetylation. Results showed a significant increase in Sin3 and HDAC1 levels in HSV-1 infected neurons 24 h p.r. with respect to matched CTRL, whereas no significant changes were detected 24 h after the primary infection (Figure 2a,b; 24 h p.r.: Sin3, * *p* = 0.017 vs. CTRL; HDAC1, * *p* = 0.035 vs. CTRL). These results were congruent with the decrease found in H3K56ac level suggesting that recurrent HSV-1 infection, and not primary infection, by affecting these protein levels, may modulate the histone acetylation ones. However, we also observed a time-dependent increase in the expression levels of both proteins, probably due to the HSV-1 latent and long-term infection.

### 2.3. Recurrent HSV-1 Infection Induces HIRA Increase in 24 h p.r. Neurons

Next, we investigated the expression levels of the senescence-associated HIRA protein. This key senescence-regulator factor was required for the formation of SAHF during cell aging. Recently, the role of HIRA in the establishment of HSV-1 latency in the PML-NB has been demonstrated [48]. Thus, we analyzed HIRA expression levels in primary neuronal culture harvested 24 h p.i. or 24 h p.r. Figure 2c shows a significant increment of HIRA following virus reactivation (** *p* = 0.009). On the contrary, no significant increase on the protein expression was observed after primary infection (24 h p.i.). First, as expected, we noticed that HIRA levels increased in control neurons within the days in culture, indicating the aging of neuronal cultures). Moreover, we observed a significant increment in HIRA during long-lasting infection (24 h p.r.) (**** *p* < 0.0001). These results indicate that HSV-1 significantly increased HIRA protein level in the long-term infection.

### 2.4. Recurrent HSV-1 Infection Modifies Histone Acetylation Profile and Sin3/HDAC1 Protein Levels in Mice

Since aging involves many mechanisms that could be minimized in a simple model as a cell culture, we deepened our study in vivo, by exploiting a mouse model of recurrent HSV-1 infection set up in our lab, that was previously virologically and molecularly characterized [3]. Specifically, we decided to analyze brain tissues from subgroups of mice (Figure 3a) sacrificed at different stages of HSV-1 infection or Mock-infection: (i) a subgroup of HSV-1-infected mice (HSV1-M) and Mock-infected mice (CTRL-M) was sacrificed 4 days after infection, during the acute phase of primary infection (i.e., when the virus was still actively replicating, and it have reached several brain areas as previously documented [3]). These mice were named hereafter HSV1-M and CTRL-M 4 d.p.i.; (ii) two subgroups of mice underwent several TS-inducing viral reactivation over life and were sacrificed following the 3rd TS (at 6 months of age, named hereafter Post 3TS) or the 7th TS (13 months of age, named hereafter Post 7TS), i.e., when the virus was actively replicating in the brain; (iii) a subgroup of mice underwent 6 TSs and was sacrificed 24 h just before the 7th TS (named hereafter Pre 7TS), that was when the virus was dormant in the brain, after 6 cycles of virus reactivation.

Firstly, we checked whether multiple cycles of viral latency and reactivation as well as primary virus infection could affect the H4K16 and H3K56 acetylation patterns in mouse brains by WB analysis of cortical homogenates from Post 3TS, Pre 7TS, Post 7TS mice, and 4 d.p.i. mice. We found that TS-induced viral replications caused a significant increase in histone H4K16 acetylation (Figure 3b) (Post 3TS: ** *p* = 0.001 vs. CTRL-M; Post 7TS: *** *p* = 0.001 vs. CTRL-M). We also detected a high level in acetylation in Pre 7TS mice, indicating that this trend was maintained within recurrent virus infection in the brain (* *p* = 0.03 vs. CTRL-M). On the contrary, we found no significant changes in 4 d.p.i. mice, in line with the data obtained with the in vitro model of HSV-1 infection following the primary infection.

Regarding the lysine 56 acetylation, we detected a significant decrease in H3K56ac levels in HSV1-M compared to those observed in CTRL-M (Figure 3c) 24 h after both the 3rd and 7th TSs, suggesting again a possible long-lasting effect of the viral infection on the acetylation of this residue (Post 3TS: ** *p* = 0.002 vs. CTRL-M; Post 7TS: * *p* = 0.033 vs. CTRL-M). Unexpectedly, the H3K56ac levels were significantly higher in Pre 7TS HSV1-M with respect to matched CTRL- M and to those of Post 7TS HSV1-M (Figure 3c, bar graphs Pre 7TS; *** *p* < 0.0001 vs. CTRL-M).

Then, to verify if multiple cycles of viral latency and reactivation could also affect Sin3 and HDAC1 levels, we investigated their protein levels in mouse cortices. We found a significant increase in Sin3 and HDAC1 protein expression levels in 4 d.p.i. HSV1-M with respect to CTRL-M (Figure 4a; 4 d.p.i.: Sin3, * *p* = 0.010 vs. CTRL-M; Figure 4b, HDAC1, * *p* = 0.046 vs. CTRL-M). Moreover, this trend was observed also after multiple viral reactivations (Figure 4a; Post 7TS Sin3, *** *p* < 0.001 vs. CTRL-M; Figure 4b, HDAC1, ** *p* = 0.008 vs. CTRL-M). The most consistent increase in Sin3/HDAC1 levels was found in Pre 7TS mice, i.e., when the virus was in the latent phase, compared to matched controls (Pre 7TS: Sin3, *** *p* < 0.0001 vs. CTRL-M; HDAC1, ** *p* = 0.006 vs. CTRL-M). Indeed, the levels of both Sin3 and HDAC1 were markedly higher than those found in mice subjected to TS-induced reactivation. These results confirm the Sin3 and HDAC1 expression observed in vitro, picturing a complex virus-host cell relationship throughout host life/time.

### 2.5. Recurrent HSV-1 Infection Induces HIRA Increase during Viral Latency in Mice

Next, we checked whether modulation of HIRA expression occurred also during the recurrent HSV-1 infection in our mouse model. In marked contrast to what we found in 24 h p.r. neurons, results showed a drastic reduction (up to 50%) in HIRA protein amounts following the 3rd TS in HSV1-M with respect to matched controls (Figure 5; bar plot; Post 3TS: *** *p* < 0.0001 vs. CTRL-M). A lower trend of decrease was also found in HSV-1-infected Post 7TS mice, suggesting an effect of viral reactivation on the protein half-life in mouse cortex. On the contrary, we detected a notable increase of HIRA levels in Pre 7TS HSV1-M, as shown in Figure 5 (bar graph; Pre 7TS: *** *p* = 0.0004 vs. CTRL-M). These levels were significantly higher than those found in both Post 3TS and Post 7TS HSV-1-infected mice. This result suggests that, after several cycles of TS, HIRA is more produced or preserved, and accumulates in cortical neurons during the viral latency. Consistently, the slight HIRA modulation found in HSV1-M Post 7TS may result from a balance between the aging-dependent accumulation of HIRA and the HSV-1-induced downregulation. Notably, no differences were observed 4 d.p.i. This probably depends on the number of neurons infected after only 4 days from primary infection and the total amount of HIRA in 6–8 weeks-old mice.

To gain insight into these results, we carried out confocal immunofluorescence analyses (IF) of mouse brain slices to investigate the localization of HIRA protein (Figure 6). IF staining was performed on brain slices obtained from CTRL-M and HSV1-M sacrificed 24 h before or after the 7th TS with a specific anti-HIRA protein antibody (green). Slices were also stained with anti-NeuN antibody (red) to mark nuclei and perinuclear cytoplasm of neurons, and diamidino-2-phenylindole (DAPI, blue) to stain all cellular nuclei. Confocal IF analysis confirmed the trend we previously observed in WB. Specifically, the mean fluorescence intensity of neuronal HIRA protein (i.e., detected in NeuN^+^ cells) was found to be significantly increased in Pre 7TS HSV1-M with respect to age-matched control mice (Figure 6; Pre 7TS ** *p* = 0.004 vs. CTRL-M). Moreover, HIRA seems to be affected by virus reactivation as its intensity was found slightly decreased in Post 7TS HSV1-M tested compared to Pre 7TS mice, although it was significantly higher with respect to matched CTRL-M (Figure 6; Post 7TS * *p* = 0.02 vs. CTRL-M). Furthermore, the nuclear localization of the protein changed in the infected Pre 7TS where HIRA was distributed perinuclearly in a more extensive way, contrary to CTRL-M, where HIRA was localized in more compact foci. Similarly, this occurred also in the Post 7TS, although the quantity of the protein decreased after reactivation.

Altogether these data indicate that HIRA is stable during viral latency, probably aiding the viral DNA repression, whereas TS-induced HSV-1 reactivation modifies the protein stability inducing its degradation and/or delocalization from the nucleus, similarly to other repressive factors (e.g., PML).

## 3. Discussion

In the present study, we addressed the hypothesis that recurrent HSV-1 infection may accelerate brain aging. To our knowledge, very few pieces of information are currently available about the appearance of aging markers and modulation of aging-related factors within recurrent HSV-1 infection. Previous studies of our group demonstrated that multiple HSV-1 reactivations in wild-type mice lead to the accumulation of neurodegenerative hallmarks and cognitive deficits typical of AD brain [3]. In this study, we exploited this experimental model and set up a compatible in vitro model of recurrent HSV-1 infection (i.e., HSV-1 infection followed by the establishment of a 10-day latency and, then, by TS-induced reactivation). Our results showed that both in vitro and in vivo recurrent HSV-1 infection modulate histone acetylation levels of two lysine residues (K16 of H4 and K56 of H3) that have been related to the aging process in several organisms. The same residues are known to be modified during acute HSV-1 infection in vitro [25,26,49,50,51]. In our study, we found that HSV-1 induced a significant increase in H4K16ac after multiple reactivations in vivo (Figure 3b), while a slight but reproducible increase was observed following a single reactivation in vitro (Figure 1c). Interestingly, H4K16ac increment was also found during viral latency in Pre 7TS mice, suggesting that recurrent HSV-1 infection may induce the persistence of higher H4K16ac levels, which is indicative of accelerated aging. Indeed, O’Sullivan (2010) reported that the increase in H4K16ac is an aging-related chromatin marker and regulates the chromatin state. Our data are in line also with those reported by Nativio et al. (2018), which showed H4K16ac increased in aged human brain samples (post-mortem) with respect to control of young brain samples [26]. On the contrary, Contrepois et al. (2012) reported a decrease in H4K16 acetylation during replicative senescence. This discrepancy may be due to the use of different experimental models: Contrepois et al. exploited human embryonic fibroblasts forced to replicative senescence via overexpression of RAF skipping the DNA damage checkpoint activation [52], whereas our data were obtained in primary wild type neurons (i.e., post-mitotic cells that undergone senescence-like phenotype) and mouse cortices from HSV-1 infected wild-type mice. Although similar pathways are involved, it is also reasonable to hypothesize that the oncogene-induced- and virus-induced-senescence may drive different effects on H4K16 residue. Moreover, it is possible that the increase of H4K16ac may be related to other factors, including the age-related decrease of Sirt1, one of the histone deacetylases responsible for this modification, which is also known to be modulated during HSV-1 infection [53,54].

Regarding H3K56 acetylation level, whose decrement was documented both in yeast and human cell cultures during replicative senescence [25,47], we found that it is not affected in HSV-1 acute infection (both in vitro and in vivo), whereas a decrease in this PTM was observed in neurons 24 h after virus reactivation from latency (Figure 1c) as well as in the mouse model of recurrent HSV-1 infection (Figure 3c). We also detected a significant H3K56ac increase in those mice analyzed before the 7th TS. Taken together, these results suggest that H3K56 deacetylation may be triggered by the productive infection that follows virus reactivation, whereas other mechanisms may be activated within the viral latency. Therefore, we argued that this modification might not correlate with aging under our experimental conditions. Interestingly, Miller and colleagues showed that H3K56 deacetylation is also a downstream DNA damage responsive PTM carried out by HDAC1, and our and other groups previously demonstrated that HSV-1 induces DNA damage during productive infection [28,55]. Hence, it is possible to speculate that the detected H3K56ac decrease is a DNA-damage indicator. Further studies are required to clarify where these markers are positioned in the chromatin landscape.

HDAC1 and Sin3 are involved in many cellular processes, including the antiviral response. Gilbert et al. showed that HDAC1 increased in human *postmortem* brains in an age-dependent manner, and Sin3 has also been associated with senescence [32,56]. In our models, we found that both Sin3 and HDAC1 protein levels raised during the recurrent HSV-1 infection (Figure 2a,b, and Figure 4) with respect to matched controls, supporting the view that multiple HSV-1 reactivations may accelerate brain aging. The increase of these protein levels was also detected during latency (in vivo, pre7TS) and may be linked to the repression of the viral genome in the PML-NB. However, we found different results in in vitro (24 h p.i.) and in vivo (4 d.p.i. mice) models of primary infection on Sin3 and HDAC1 levels. It is possible that HSV-1-induced Sin3/HDAC1 increase in vivo may be related to inflammatory response due to viral infection, whereas in vitro neurons lacking their immune-related counterpart (glial cells) did not show this mechanism.

Next, we focused our attention on HIRA protein expression and localization. HIRA complex colocalizes with the PML-NBs in senescent cells, and during HSV-1 latent infection, HIRA is involved in the repression of the viral genome. Recently, Rai et al. demonstrated that H4K16ac is enhanced on the promoters of expressed genes in senescent cells, and its enrichment is preserved by HIRA protein [57]. Our in vitro results showed that HIRA expression levels significantly increased in 24 h p.r. infected neurons with respect to matched CTRL and 24 h p.i. neurons (Figure 2c), suggesting that a cycle of HSV-1 latency and reactivation may contribute to accelerate aging. In addition, HIRA levels also increased in control neurons within the days in culture, confirming that the expression of this protein normally increases during aging. Moreover, we did not observe any significant change in HIRA levels 24 h p.i. This latter result is in line with McFarlane et al. showing that ICP0 disrupts nuclear localization of HIRA without inducing its proteasome-dependent degradation in several non-neuronal cell lines during HSV-1 productive infection [41]. However, in mouse brains, we found that 3 cycles of viral reactivation resulted in a significant decrease of HIRA expression (Figure 5), whereas a slight decrease was observed following 7 reactivations (Figure 5 and Figure 6). On the contrary, in Pre 7TS HSV1-M brains HIRA was found significantly increased with respect to CTRL-M. This discrepancy between the in vitro and in vivo results may be due to the in vitro model limitations: the number of reactivations induced (1 TS vs. 3 TS/7 TS) could also influence the viral copy number/viral load causing a more dramatic decrease in HIRA levels in vivo and the in vitro latency establishment. Indeed, a recent study by Powell-Doherty and colleagues has shown that ACV administration on hippocampal neuronal culture induced a “quiescent infection”, just preventing viral replication instead of an actual latent state establishment. In this perspective, it is possible to argue that the HIRA increase in 24 p.r. infected neurons is a result of an accumulation of HIRA due to the maintenance of the quiescent state [58]. Taken together, these results showed that recurrent HSV-1 infection induces the degradation of HIRA protein only within the active replication of the virus, whereas during latency, the expression of the protein is upregulated. It is noteworthy that a similar trend was also observed for H3K56ac, which is known to be induced by ASF1, a cellular protein associated with HIRA only during senescence. It is possible to speculate that the increase of HIRA may hijack ASF1 to the SAHF/PML-NB, indirectly inducing the increase of H3K56ac that we found in Pre 7TS mice when the virus is still latent and is localized in PML bodies. Notable, in HSV1-M brains HIRA was found localized perinuclearly, where it probably also retains the viral genome and the PML-NB (Figure 6).

In summary, our data indicate that HSV-1 harboring neurons may accumulate markers of senescence such as H4K16 hyperacetylation and HIRA protein, together with the increase in Sin3/HDAC1, that may accelerate neuronal aging. Altogether these results strongly suggest that recurrent HSV-1 infection can accelerate pathological aging of neurons through the modulation/regulation of these proteins/mechanisms thus contributing to neurodegeneration.

## 4. Materials and Methods

### 4.1. Virus Production and Titration

HSV-1 (F strain, wild-type) production was performed on monolayers of African green monkey kidney cells (VERO, ATCC) cultivated in 75 cm^2^ tissue culture flasks and infected a multiplicity of infection (M.O.I.) of 0.01. After 48 h at 37 °C, HSV-1-infected cells were harvested, and following 3 freeze/thaw cycles, cell debris were removed with low-speed centrifugation. Then the supernatant containing new virions was collected and titred by Standard Plaque Assay (SPA) as plaque-forming unit (PFU) per ml (PFU/mL) on confluent VERO monolayer in 24-wells plates [59]. Briefly, after the adsorption phase (1 h, 37 °C), the supernatants were discarded, and the medium was replaced with RPMI medium (Sigma-Aldrich, St Louis, MO, USA) supplemented with 2% heat-inactivated Fetal Bovine Serum (FBS, Gibco, Thermo Fisher Scientific, Waltham, MA, USA) and 2% carboxymethyl cellulose (CMC, Sigma-Aldrich, St Louis, MO, USA), to contain the virus infection. After 48 h, VERO was subjected to a washing-step in Dulbecco’s Phosphate Buffered Saline (PBS, w/o Calcium chloride and Magnesium chloride, Sigma-Aldrich, St Louis, MO, USA), fixed in absolute methanol and colored with crystal violet (Sigma-Aldrich, St Louis, MO, USA) to evaluate the plaque formation. In this study, the virus had a titer of 1 × 10^9^ PFU/mL. Similarly, a Mock-solution (Mock) was prepared from not infected VERO cells.

### 4.2. Neuronal in Vitro Model of Recurrent HSV-1 Infection

Primary cultures of cortical neurons were prepared from E17 WISTAR rat embryos (Charles Rivers Laboratories International, Inc., Wilmington, MA, US), as described in [46,55] with minor modification. Briefly, embryos were decapitated to separate the head from the rest of the body, then the brains were dissected to isolate the cortices. Cortices deprived of meninges were collected in cold PBS, and then incubated for 15 min at 37 °C in trypsin-EDTA (0.025%/0.01% *w*/*v*; 500 µL/cortices pair; Gibco, Thermo Fisher Scientific, Waltham, MA, USA). After blocking the trypsin with FBS, the tissues were mechanically dissociated at room temperature (RT) with a sterilized Pasteur pipette, and the cell suspension was centrifuged (1200 rpm, 3 min). The pellet was suspended in complete MEM medium (5% FBS, 5% horse serum, 1% glutamine (2 mM), 1% penicillin-streptomycin-neomycin antibiotic cocktail (PSN), and glucose (25 mM); Sigma- Aldrich, St Louis, MO, USA). Cells were plated at a density of 1 × 10^6^ cells on precoated 6 well-plates (coating: poly-L-lysine 0.1 mg/mL; Sigma-Aldrich, St Louis, MO, USA) and kept in a 37 °C, 5% CO_2_ humidified incubator. After 4 hours, the culture medium was replaced with Neurobasal medium (Gibco, Thermo Fisher Scientific, Waltham, MA, USA) containing 2% B-27 (Invitrogen, Carlsbad, CA, USA), 0.5% glutamine (2 mM), and 1% PSN. The day after (day in vitro 1, DIV 1), the medium was supplemented with 5 μM cytosine arabinoside (Ara-C, Sigma-Aldrich, St Louis, MO, USA) to prevent glial proliferation and obtain a pure neuronal culture. Four days after plating (DIV 4), the medium was replaced with glutamine-free Neurobasal medium supplemented with 2% B-27 and 1% PSN. Neurons were cultured for 6–7 days, with half the medium refreshed every 48 h, before experiments could be carried out.

At DIV 6, except for an untreated maintenance plate, neuronal cultures were separated into 2 main experimental groups: 24 h post-infection (p.i.) and 24 h post reactivation (24 h p.r.; 11 days p.i.). The medium of the second group was supplemented with 50 µM acycloguanosine (ACV, Sigma-Aldrich, St Louis, MO, USA) for 18 h before infection (37 °C, 5% CO_2_). At DIV 7, the neuronal cultures were Mock- or HSV-1-infected (0.1 M.O.I.) in the Neurobasal medium (adsorption phase; 1 h and 30 min, 37 °C). Then, the medium was removed and, after a wash in PBS. The cells were returned to the original medium (+/−ACV) and cultured for 24 h or 11 days. The 24 h p.i. neurons were harvested and collected for protein extraction (see below). The 24 h p.r. neurons medium was refreshed every 2–3 days and supplemented with 50 µM ACV, to keep the virus in the latent state in the HSV-1-infected one at DIV 17 (d.p.i. 10), the medium was replaced to ensure ACV depletion with half of conditioned-Neurobasal medium, obtained from the matched maintenance plate, and half of fresh medium. Then, the cultures were subjected to a Thermal Stress (TS) cycle to induce viral reactivation by using a constant-temperature water bath set at 43 °C for 10 min. Mock-infected neurons underwent a superimposable protocol (ACV treatment and TS). No sign of neuronal death or distress was observed after this procedure. A total of 24 hours later, the neurons were harvested. The supernatants were collected 24 h p.i., 24 h p.r., and at every medium-refresh step and analyzed by SPA, to evaluate the viral titer and the viral latency/reactivation. Supernatants from representative experiments were also analyzed by In Cell Western (ICW) to quantify virus titer (Appendix A), as described in [60].

### 4.3. Mouse Model of Recurrent HSV-1 Infection and Experimental Groups

To investigate the aging markers, we used the brain tissues from the mouse model of recurrent HSV-1 infection previously set up and virologically characterized in our group, as thoroughly described in De Chiara et al. 2019 [3]. Briefly, 6–8-week-old female BALB/c mice had been housed in the animal house (Istituto Superiore di Sanità, Rome, Italy) and has been kept under a 12 h light/dark cycle at RT with free access to food and water. A group of mice was HSV-1- inoculated (suspension equivalent to 10^6^ PFU/mL) by snout abrasion to establish the first experimental group named HSV1-M, following anesthesia with a Ketamine (80 mg/Kg)/Xylazine (5 mg/Kg) cocktail intraperitoneally injected. At the same time, a set of mice was analogously inoculated with the Mock-solution constituting the CTRL-M group and was kept in separated cages. A first group of mice was sacrificed 4 days post-infection (4 d.p.i., i.e., when the virus actively replicates), whereas, after the establishment of latent infection (6 weeks p.i.), the other mice underwent to repeated cycles of TS throughout their life (up to 7 times) to induce viral reactivation. Then, mice were analyzed at specific timepoints: (i) at 4 d.p.i (CTRL-M *n* = 3 and HSV1-M *n* = 5) (ii) after the 3rd TS (Post 3TS, CTRL-M *n* = 3 and HSV1-M *n* = 4; 6-months-old); (iii) 24 h before the 7th TS (Pre 7TS, CTRL-M *n* = 4 and HSV1-M *n* = 4; 13-months-old); (iv) 24 h after the 7th TS (Post 7TS, CTRL-M *n* = 4 and HSV1-M *n* = 4; same age of the Pre 7TS).

### 4.4. Ethics Statement

All the experimental protocols of the present study were in compliance with the European Guide for the Care and Use of Laboratory Animals and institutional guidelines and with the Italian legislation on animal experimentation (Decreto legislativo n. 26/2014, Direttiva UE 63/2010) to avoid any unnecessary distress during the experimental procedures. The experimental protocols were reviewed by the Animal Welfare Body (Istituto Superiore Sanità) and authorized by the Italian Ministry of Health (protocol numbers 801/2016-B, 745/2016-PR).

### 4.5. Western Blotting

Neuronal cultures were harvested and lysed in 300 µL RIPA buffer (20 mM Tris, 150 mM NaCl, 1% Triton X-100, 1% sodium deoxycholate, 0.1% SDS) containing 5 µM Sodium Butyrate and protease and phosphatase inhibitors cocktails (Sigma-Aldrich, St Louis, MO, USA). Cortices, after dissection, were stored at −80 °C until use and were homogenized on ice using a potter (15–25 strokes), in 5 volumes of complete RIPA. The neuronal and cortical proteins in RIPA were sonicated at low power setting on ice (3 pulse × 3 s), and then separated by centrifugation at 14,000 rpm at 4 °C for 20 min. Total protein concentrations were measured with Micro BCA method (Thermo Fisher Scientific, Waltham, MA, USA). An equal amount of protein samples (30 μg) was separated by 7.6–15% SDS-PAGE and blotted onto 0.22/0.45 μm nitrocellulose membrane (Bio-Rad, Hercules, CA, USA; GE Healthcare, Cytiva, Marlborough, MA, USA). Membranes were colored with Ponceau Red (Sigma-Aldrich, St Louis, MO, USA) and, after washing with TBS, blocked in 10% non-fat milk (1–3 h), in 0.1% Tween-20 TBS (T-TBS). Then the filters were incubated overnight at 4 °C with the primary antibody (Ab) diluted in 5% non-fat milk in T-TBS. We used the following dilution for primary Abs: Anti-HIRA 1:1000 (#12463, Cell Signaling Technology, Danvers, MA, USA),), Anti-H3K56ac 1:1000 (#4243S, Cell Signaling Technology, Danvers, MA, USA), Anti-H4K16ac 1:1000 (#13534, Cell Signaling Technology, Danvers, MA, USA); Anti-H3 1:2500 (#1791, Abcam, Cambridge, UK); Anti-Sin3 1:200 (sc-5299, Santa-Cruz, Biotechnology, Heidelberg, Germany), Anti-HDAC1 1:1000 (sc-81598, Santa-Cruz, Biotechnology, Heidelberg, Germany); Anti- β-Actin 1:5000 (A2228, Sigma-Aldrich, St Louis, MO, USA). After 3 × 10 min washes with T-TBS, the membranes and incubated 1 h with secondary Ab (HRP-conjugated antibodies, Jackson ImmunoResearch Laboratories, West Grove, PA, USA), followed by 3 × 10 min washes in T-TBS. The chemiluminescence reaction was obtained using Clarity or Clarity Max Western ECL Substrate (Bio-Rad, Hercules, CA, USA) and detected by film impression or Chemidoc Imaging System (Bio-Rad, Hercules, CA, USA). Densitometric analysis was performed using Quantity One software or ImageLab software (Bio-Rad, Hercules, CA, USA). The integrated density, corrected for non-specific background, was normalized to equal loading housekeeping protein (β-Actin, or H3) and averaged per animal for mice analyses. These values were included in the statistical analysis. Finally, we chose histone H3 as load control for all the analyzed histones PTMs since H3 and H4 are obligate heterodimers and are stoichiometrically equal in chromatin occupancy.

### 4.6. Immunofluorescence Analysis

Slices samples used were previously obtained from our group [3]. Briefly, after anesthesia with an intraperitoneal injection of a Ketamine (200 mg/Kg) and Xylazine (10 mg/Kg) cocktail, each mouse was immobilized and transcardially perfused with PBS (pH 7.4), and with 4% paraformaldehyde (PFA). The brains were fixed for 24 h at 4 °C in PFA, followed by cryoprotection in 15–30% sucrose gradient in PBS. A total of 40-μm-thick coronal slices were cut and stored at −20 °C in cryoprotectant solution (pH 7.4 50 mM sodium phosphate buffer, 15% glucose, 30% ethylene glycol) until use. For this study, 2 slices for each mouse were used, washed in PBS and permeabilized (0.5% Triton X-100 in PBS) for 15 min at RT. After the samples were blocked with blocking solution (10% horse serum, 0.2% Triton X-100 in PBS, RT) for 90 min to avoid unspecific Abs binding, the slices were incubated overnight with the chosen primary Ab (5% horse serum, 0.2% Triton X-100 PBS, +4 °C). The following dilutions were used: 1:200 anti-HIRA Ab (#12463, Cell Signaling Technology, Danvers, MA, USA); 1:300 anti-NeuN Ab (MAB377, SantaCruz Biotechnology, Heidelberg, Germany). Slices undergone to 3 washing steps (PBS), and incubated for 90 min at RT with specific secondary Abs coupled to Alexa 488 or Alexa 546 (1:1000, Invitrogen). After 3 washes in PBS, slices were stained with DAPI (1:500 dilution in PBS) for 20 min at RT, briefly washed, dried, and then mounted on a coverslip with mounting medium (ProLong Gold Anti-fade reagent, Invitrogen, Carlsbad, CA, USA). Images were acquired at 40× magnification with a confocal laser scanning system (Leica Microsystems TCS-SP2, Wetzlar, Germany) and an oil-immersion objective (N.A. 1.4). Quantifications of image intensity were carried out by Imaris suite 7.4 software (Bitplane A.G., Zurich, Switzerland) in neocortical areas of the slices. The “Surface module” was used to automatically draw a mask on NeuN positive cells (red channel) to identify neurons, and then the mean pixel intensity of HIRA (green channel) was evaluated in neuronal cells. For inset images, an additional 10× magnification was applied.

### 4.7. Statistical Analyses

Data were analyzed by Mann–Whitney test, multiple Student’s *t*-test, two-way ANOVA model, when appropriate, using the statistical software Graphpad Prism v. 6.0. Experiments were repeated up to 3 times. Post-hoc comparisons were performed using Tukey’s honest significance test [61] and outliers were identified by using the Grubbs test and omitted from the analyses [62]. Data were presented as mean ± standard error of the mean (SEM). The level of significance (alpha) was set at 0.05.

## Figures and Tables

**Figure 1 ijms-22-06279-f001:**
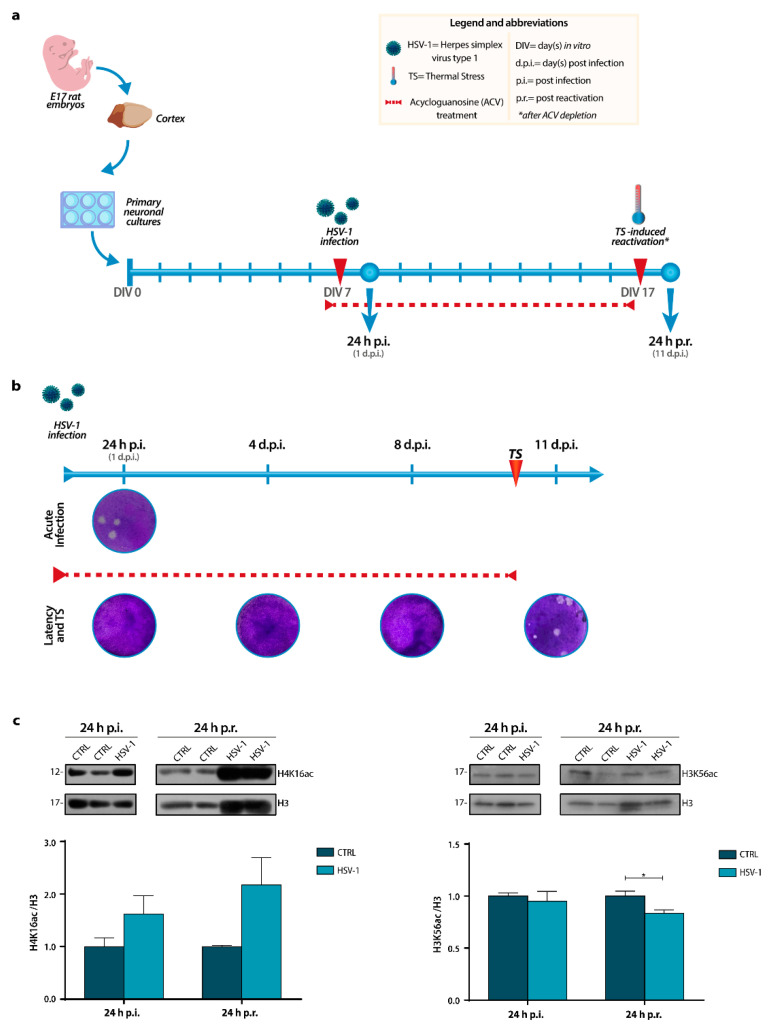
In vitro experimental model and histones acetylation levels during a recurrent HSV-1 infection in neurons. (**a**) Timeline of experimental procedures showing the days in culture of the neurons, the ACV treatment (red dashed line), the time of primary infection (DIV 7) and reactivation (DIV 17), and the time points chosen for the analyses: 24 h p.i. (DIV 8) and 24 h p.r. (DIV18). Abbreviations are shown in the yellow box. (**b**) Representative images of standard plaque assay (SPA) after primary infection (24 h p.i.) with or without ACV (Acute Infection and Latency and TS rows, respectively), during ACV-induced latency (4 and 8 d.p.i.) or after TS-induced reactivation (11 d.p.i.). (**c**) Representative immunoblots showing H4K16 and H3K56 acetylation levels in lysates from Mock- and HSV-1-infected neuronal cultures harvested 24 h p.i. or 24 h p.r.. H3 staining was used as a loading control. Densitometric analyses were performed with Quantity One software and normalized to H3 expression. The values represent the normalized fold-changes in acetylation levels from HSV-1 samples with respect to matched CTRL. Data are mean ± SEM. * *p* < 0.05. Statistical significance was calculated by the Mann–Whitney test.

**Figure 2 ijms-22-06279-f002:**
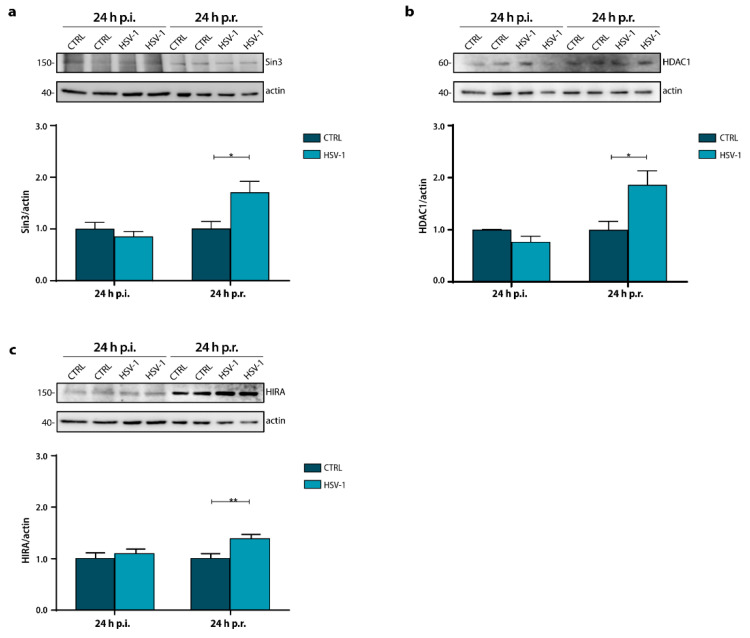
Sin3, HDAC1 and HIRA expression during a recurrent HSV-1 infection in vitro. Representative immunoblots showing Sin3 (**a**), HDAC1 (**b**), and HIRA (**c**) protein levels in lysates from Mock- and HSV-1-infected neuronal cultures harvested 24 h p.i. or 24 h p.r. Actin staining was used as a loading control. Densitometric analyses were performed with ImageLab software and normalized to actin expression. The values represent the normalized fold-changes in protein levels from 24 h p.r. CTRL or HSV-1 samples with respect to 24 h p.i. CTRL. Data are mean ± SEM of three independent experiments. * *p* < 0.05; ** *p* < 0.01. Statistical significance was calculated by Mann–Whitney test.

**Figure 3 ijms-22-06279-f003:**
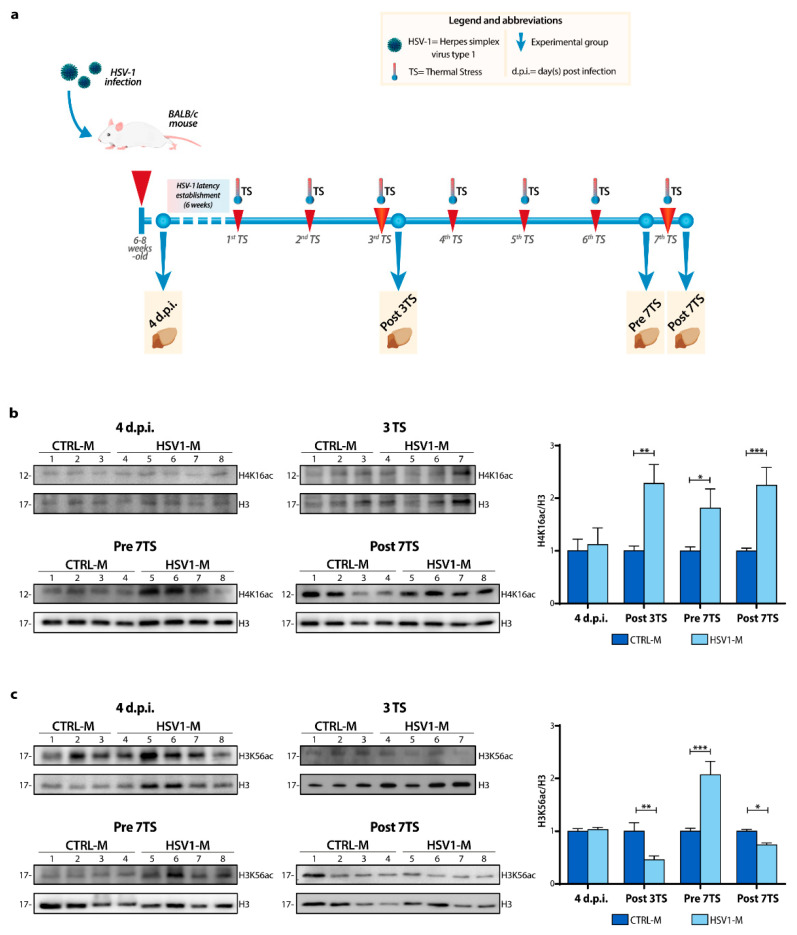
Schematic representation of the recurrent HSV-1 infection and modulation of histone acetylation levels in vivo. (**a**) Timeline showing HSV-1 infection, TS-induced virus reactivations, and chosen experimental time-points over the mouse life. The HSV-1 inoculation (primary infection) was performed by snout abrasion (first red arrow). TS induces viral reactivations are indicated by red arrows. The dashed line indicates the establishment of the latent infection. Mice were sacrificed and analyzed at the following time points (blue dots) indicated by the yellow boxes: 4 d.p.i., following the 3rd (at 6 months of age, Post 3TS) and the 7th TS (13 months of age, Post 7TS) or 24 h just before the 7th TS (Pre 7TS). Representative WBs showing H4K16 (**b**) and H3K56 (**c**) acetylation levels in the whole lysate from the cortices of 4 d.p.i., Post 3TS, Pre and Post 7TS mice. Densitometric analyses were performed with ImageLab software and normalized to H3 or tubulin expression. The values represent the normalized fold-changes in protein levels from HSV1-M with respect to CTRL-M. Data are mean ± SEM. * *p* < 0.05; ** *p* < 0.01; *** *p* < 0.001. Statistical significance was calculated by multiple t-test and determined using the Sidak–Bonferroni method, with alpha = 0.05.

**Figure 4 ijms-22-06279-f004:**
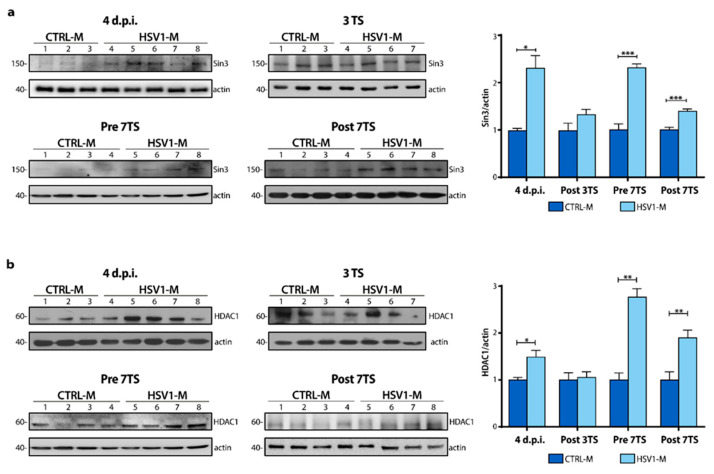
Recurrent HSV-1 infection induces an increase in Sin3 and HDAC1 protein expression. Representative WBs showing Sin3 (**a**) and HDAC1 (**b**) protein levels in the whole lysate from the cortices of 4 d.p.i., Post 3TS, Pre and Post 7TS mice. Densitometric analyses were performed with ImageLab software and normalized to H3 or tubulin expression. The values represent the normalized fold-changes in protein levels from HSV1-M with respect to CTRL-M. Data are mean ± SEM. * *p* < 0.05; ** *p* < 0.01; *** *p* < 0.001. Statistical significance was calculated by multiple *t*-tests and determined using the Sidak–Bonferroni method, with alpha = 0.05.

**Figure 5 ijms-22-06279-f005:**
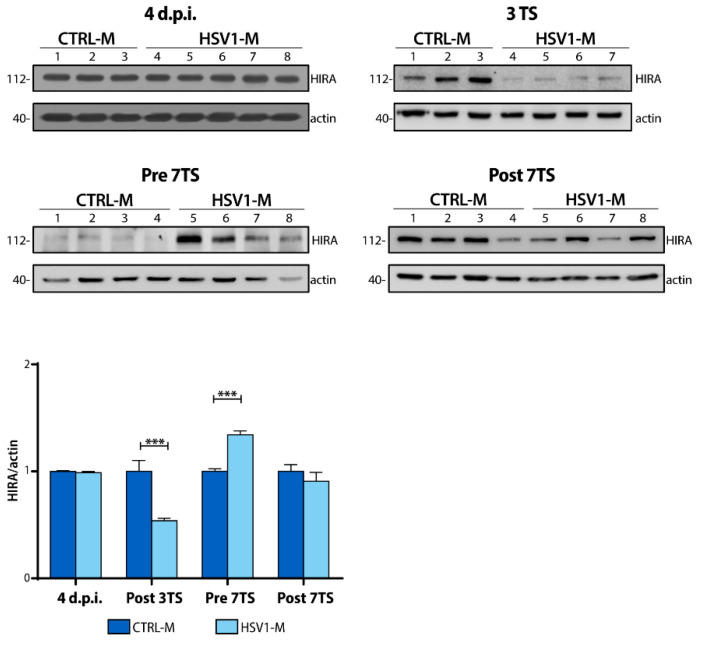
HIRA protein expression in the cortex of recurrent HSV-1 infection mouse model. Representative WBs of HIRA protein expression 4 d.p.i., 24 h after the 3rd and the 7th TS (Post 3TS and Post 7TS respectively), or 24 h before the 7th TS (Pre 7TS). Densitometric analyses were performed with ImageLab software and normalized to actin expression level. The values represent the normalized fold-changes in protein levels from HSV1-M with respect to CTRL-M (bar graph on the left). Data are mean ± SEM. *** *p* < 0.001. Statistical significance was calculated by multiple t-tests and determined using the Sidak–Bonferroni method (bar graph, alpha = 0.05).

**Figure 6 ijms-22-06279-f006:**
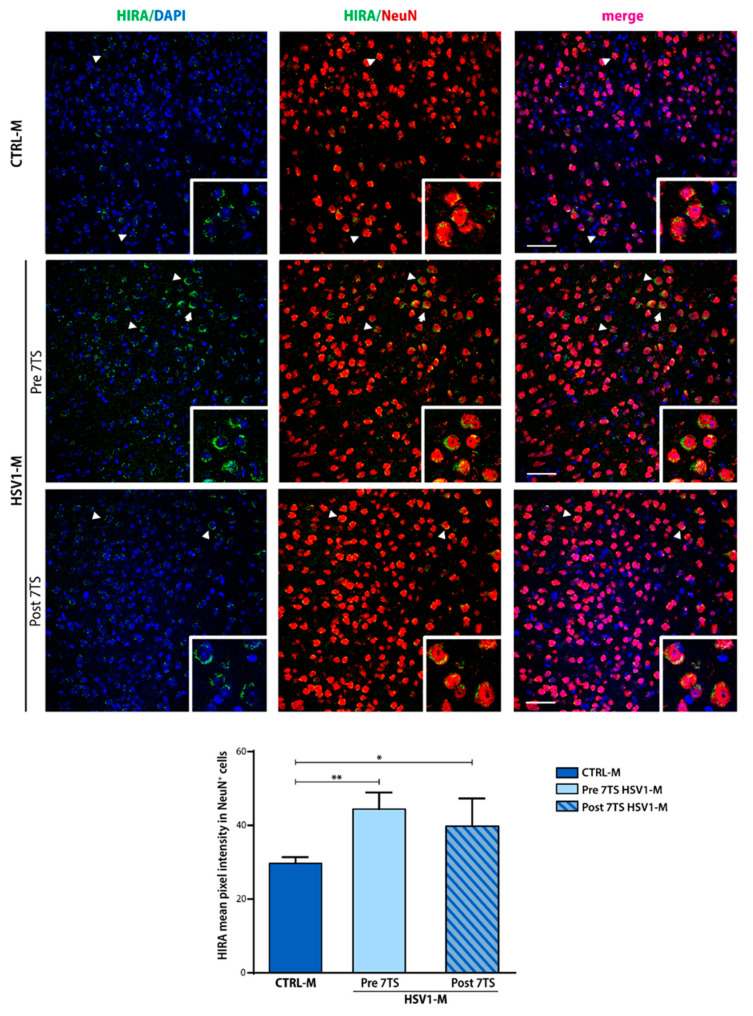
Cortical HIRA intensity changes during the recurrent HSV-1 infection. Confocal immunofluorescence analysis showing the immunostaining for NeuN (red) and HIRA (green) in coronal brain slices from HSV-1- or Mock-infected mice (HSV1-M and CTRL-M, respectively) sacrificed before or after the 7th TS (13 months of age). Cellular nuclei are stained by DAPI (blue). The insets represent the enlargement of HIRA-positive neurons pointed by white arrowheads (10× magnification). Bar graphs show the HIRA mean pixel intensity quantified in NeuN-positive cells (NeuN^+^, i.e., neurons). Two coronal sections were analyzed for each brain. * *p* < 0.05; ** *p* < 0.01. Statistical significance was calculated by multiple t-tests and determined using the Sidak–Bonferroni method (bar graph, alpha = 0.05). Scale bar: 50 µm.

## Data Availability

All relevant data are within the manuscript and its Appendix A.

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
