# Peer review of "Recurrent Herpes Simplex Virus Type 1 (HSV-1) Infection Modulates Neuronal Aging Marks in In Vitro and In Vivo Models"

_ijms, 2021, doi:10.3390/ijms22126279_

Round 1

Reviewer 1 Report

This manuscript by Napoletani et. al. studied the modulation of neuronal aging markers upon HSV-1 reactivation using in vitro and in vivo models. They focused on the changes of H3K56ac, H4K16ac, HIRA, Sin3, HDAC1. The methods they used are mostly western blots with some immunofluorescent analyses. The in vitro results showed, as described in their hypotheses, that H3K56 and H4K16 acetylation levels are contrariwise altered by HSV-1 reactivation. Sin3, HIRA, and HDAC1 all increase after reactivation. The in vivo data mostly agreed with their in vitro results with some differences. For example, their Pre 7TS samples showed significant increase of H3K56ac followed by a decrease in Post 7TS. HIRA, however, exhibited a decrease in Post 3TS then increased in Pre 7TS.

The method is straightforward and the statistics is sound. The manuscript is well-written with a good discussion. I have three minor suggestions:

  • Evidence of latency and reactivation in primary neuronal cultures: As we know, the HSV-1 latency in vitro can be tricky. We don’t know the ratio of the cells established quiescent state of infection. More importantly, the rate of reactivation varied after the ACV washout. It’d be nice to include some evidences such as viral gene expression (ICP0, TK, and LAT) studies before and after the thermal stress. In addition, the plaque assays from the Fig. 1b was not convincing. A quantitative assay using cultured media and/or the cell extract would unveil the degrees of reactivation.
  • Effects of thermal stress on animals: The in vivo model required multiple thermal stress on the animals. There is again no evidence supporting the establishment of latency and reactivation. I also wonder if the repetitive stress would kill the animals?
  • 3a was not clear: My first look at this figure gave me an impression that the first TS occurred at 5 dpi. It appeared in the materials and methods that the TS did not happen until 6 weeks after the initial infection, while latency is established. It’d be a good idea to relabel the figure.

Author Response

Reviewer 1

Comments and Suggestions for Authors

This manuscript by Napoletani et. al. studied the modulation of neuronal aging markers upon HSV-1 reactivation using in vitro and in vivo models. They focused on the changes of H3K56ac, H4K16ac, HIRA, Sin3, HDAC1. The methods they used are mostly western blots with some immunofluorescent analyses. The in vitro results showed, as described in their hypotheses, that H3K56 and H4K16 acetylation levels are contrariwise altered by HSV-1 reactivation. Sin3, HIRA, and HDAC1 all increase after reactivation. The in vivo data mostly agreed with their in vitro results with some differences. For example, their Pre 7TS samples showed significant increase of H3K56ac followed by a decrease in Post 7TS. HIRA, however, exhibited a decrease in Post 3TS then increased in Pre 7TS.

The method is straightforward and the statistics is sound. The manuscript is well-written with a good discussion.

We are grateful to the reviewer for his/her positive comments on our manuscript and careful reading

I have three minor suggestions:

Evidence of latency and reactivation in primary neuronal cultures: As we know, the HSV-1 latency in vitro can be tricky. We don’t know the ratio of the cells established quiescent state of infection. More importantly, the rate of reactivation varied after the ACV washout. It’d be nice to include some evidences such as viral gene expression (ICP0, TK, and LAT) studies before and after the thermal stress. In addition, the plaque assays from the Fig. 1b was not convincing. A quantitative assay using cultured media and/or the cell extract would unveil the degrees of reactivation.

We agree with these observations and are aware that the establishment of HSV-1 latency in vitro can be very difficult to achieve. Actually, we detected some spontaneous HSV-1 reactivations within the in vitro protocol schedule, but we did not include these experiments in our data analyses. As the referee stated, the rate of reactivation can vary after the ACV washout and we also know that virus reactivation can be dependent on ACV washout itself. However, we believe that these cases cannot affect our experimental design and conclusions. To support our idea and address referee’s concern about SPA in Figure 1, we performed new experiments quantifying virus titer in cell supernatants from one representative experiment by SPA and ICW assay (based on immunofluocescence detection/quantification of gB protein in HSV-1-infected cells, Fabiani et al 2017). Results shown in Supplementary Fig 2 indicate that, under our experimental conditions, latent infection occurred within ACV treatment as well as virus reactivation upon ACV depletion and TS. Moreover, In the revised manuscript, we replaced panel b in Figure 1 providing plaque from another representative experiment.

Effects of thermal stress on animals: The in vivo model required multiple thermal stress on the animals. There is again no evidence supporting the establishment of latency and reactivation. I also wonder if the repetitive stress would kill the animals?

A detailed description of the infection-reactivation protocol we applied to mice as well as the virological characterization of the recurrent model of HSV-1 infection is provided in Plos Pathogens 2019 (De Chiara et al), as stated in Methods. Six weeks after primary infection, mice were subjected to TS to reactivate latent virus and then TS was repeated up to seven times at 6–8 weeks of interval. Mock-infected mice underwent superimposable infection-reactivation protocol. Under our experimental conditions, low mortality rate in mice (8–12%) within the 11-month protocol schedule was observed. In this study, we analysed the cortex of mice sacrificed 4 days post infection (p.i.), after 3 and 7 Thermal stress-inducing viral reactivation (post 3TS and post 7TS, respectively), and just before the 7th TS (pre 7Ts). HSV-1 presence in the analysed tissues was checked by WB analyses of viral protein expression, PCR amplification of viral TK gene. Matched tissues from Mock-infected mice were used as controls in these analyses. Viral reactivation was checked by assessing the presence of ICP4 mRNA. We also checked the presence of infectious virus in brain tissues to assess the efficacy of TS-inducing viral reactivation. Specifically, we assessed the presence and the titer of infectious virus in brain homogenates by cell culture method (e.g, direct infection on VERO cell and In Cell Western analyses of viral protein expression). In parallel, another subset of animals was analysed by IF for gB expression. In brain tissues from pre7TS mice (where HSV-1 latent infection should be established following the 6th TS) we did not detect neither gB expression or infectious virus. All these results are shown in Plos Pathogens 2019.

3a was not clear: My first look at this figure gave me an impression that the first TS occurred at 5 dpi. It appeared in the materials and methods that the TS did not happen until 6 weeks after the initial infection, while latency is established. It’d be a good idea to relabel the figure.

We agree with the referee observation and relabelled the figure as suggested.

Reviewer 2 Report

The manuscript “Recurrent herpes simplex virus type 1 (HSV-1) infection modulates neuronal aging marks in in vitro and in vivo models” by Napoletani et al. have attempted to explore the changes resulting from latent HSV-1 infection in neuronal cells. There is a great interest in understanding these changes as latent HSV-1 infection was suggested to be linked to the development of age-related disorders such as Alzheimer’s disease. This work present evidence both in vitro and in vivo that suggest that markers linked to neuronal aging can be detected in cells infected latently with HSV-1 (following reactivation). While the authors have carried out a tremendous amount of work and provided important findings, I feel that these experiments lack essential controls required for their main conclusions.

Few major changes are required:

  1. The in vitro experiments presented in figure 1 and 2 require a non-reactivated control, IE, neuronal cells infected for a week in the presence of ACV without TS treatment. Similarly, non-infected neuronal cells treated with ACV are also missing from the analysis.
  2. To conclude that “Recurrent HSV-1 infection modulates neuronal aging marks”, the in vivo experiments require to have mice infected without several reactivations episodes to be compared with the ones that had several reactivations.
  3. There are several major differences in the amount of control proteins in the western blot panels (especially in figure 1 C, but elsewhere as well) that suggest that simple image analysis might be inaccurate due to over exposures issues.

Major changes that will highly benefit the paper but are not essential:

  1. The IF presented would benefit from identification of the cells that indeed latent, this can be done with RNA FISH for the LAT or DNA FISH for the viral genomes. This will allow the authors to conclude if the latent neurons are the neurons overgoing the changes in protein levels. IF+FISH of the in vitro analysis can also improve the understanding of the described processes.
  2. in the in vitro experiments, all the data is normalized to control at 24HPI, however in the in vivo experiments each experiment is normalized to the its own control (for example in figure 3C H3K56ac Pre7TS control is 1 although it is clearly much lower than the levels in the control at 4DPI), this result in discrepancy in the way the analysis is presented and effect the visualized interpretation of the results.

Round 2

Reviewer 2 Report

I feel the authors have not answered any of my concerns regarding the missing controls.